# Genome-Wide Analysis of the *WOX* Transcription Factor Genes in *Dendrobium catenatum* Lindl.

**DOI:** 10.3390/genes13081481

**Published:** 2022-08-19

**Authors:** Hefan Li, Cheng Li, Yuhua Wang, Xiangshi Qin, Lihua Meng, Xudong Sun

**Affiliations:** 1Key Laboratory of Yunnan for Biomass Energy and Biotechnology of Environment, School of Life Sciences, Yunnan Normal University, Kunming 650500, China; 2The Germplasm Bank of Wild Species, Kunming Institute of Botany, Chinese Academy of Sciences, Kunming 650201, China; 3University of Chinese Academy of Sciences, Beijing 100049, China

**Keywords:** *Dendrobium catenatum* Lindl., WOX transcription factor family, phylogenetic analysis, regulation of gene expression, expression profiles, phytohormone response

## Abstract

The WUSCHEL-related homeobox (WOX) proteins are a class of transcription factors exclusive to plants. They can promote cell division or inhibit stem cell differentiation to regulate plant growth and development. However, the *WOX* transcription factor genes in the monocotyledon *Dendrobium catenatum* Lindl. remain relatively uncharacterized. Specifically, the effects of phytohormones on their expression levels are unclear. In this study, we identified and analyzed 10 candidate *DcaWOX* transcription factor genes in *D. catenatum*. The *DcaWOX* family was divided into the modern/WUS, intermediate, and ancient clades. The subcellular localization analysis detected DcaWOX-GFP fusion proteins in the tobacco epidermal leaf cell nucleus. In DcaWOX, members of the WUS clade with the WUS-box motif can significantly activate the expression of *TPL* in vivo, while members of the intermediate and ancient clades cannot. The expression of the *DcaWOX* genes varied among the examined tissues. Moreover, the *DcaWOX* expression patterns were differentially affected by the phytohormone treatments, with differences detected even between homologs of the same gene. Furthermore, the gene expression patterns were consistent with the predicted cis-acting elements in the promoters. The above results suggest that *DcaWOX* may have an important role in its growth and development and resistance to stress. The results of this comprehensive investigation of the *DcaWOX* gene family provide the basis for future studies on the roles of *WOX* genes in *D. catenatum*.

## 1. Introduction

In 2004, Haecher et al. identified 14 WUS homologs in *Arabidopsis thaliana* and named them WUSCHEL-related homeobox (WOX) genes [1]. Plant-specific WOX transcription factors contain a conserved “helix-loop-helix-turn-helix” homeodomain composed of 60–66 amino acid residues [2,3]. The WOX transcription factor genes, which are highly expressed in actively growing plant tissues, affect various plant developmental stages by mediating the division of diverse meristematic cells, the development of major organs, the formation of plant embryos, and the transduction of phytohormone signals [4]. Based on the phylogenetic relationships, the WOX family members in plants have been separated into the following three clades: the ancient clade (in all plants and green algae), the intermediate clade (originated in vascular plants), and the modern clade (exclusive to spermatophytes) [5]. In *A. thaliana*, the WUS clade (i.e., modern clade) has eight members (AtWUS and AtWOX1–7), the intermediate clade consists of four members (WOX8, WOX9, WOX11, and WOX12), and the ancient clade comprises three members (WOX10, WOX13, and WOX14) [6]. Unlike the members of the other two clades, the WUS clade members have a conserved WUS-box motif. A previous study revealed that AtWUS, which includes a WUS-box motif, plays a vital role in the development of the bud tip meristem (i.e., shoot apical meristem) [7].

The WOX transcription factor genes have been identified in many plants, including *A. thaliana* (15 members) [8]; *Sorghum bicolor* (11 members) and *Zea mays* (21 members) [9]; *Salix suchowensis* (15 members) [10]; *Gossypium arboreum* (21 members), *G. raimondii* (20 members), and *G. hirsutum* (38 members) [11]; *Solanum lycopersicum* (10 members) [12]; and *Selaginella kraussiana* (8 members) [13]. The WOX genes modulate many important stages of plant growth and development by maintaining an appropriate balance between stem cell division and differentiation via the WUSCHEL–CLAVATA negative feedback loop, which is critical for flower and embryo development [14]. Notably, the WUS transcription factor controls stem cell division and differentiation by regulating cytokinin and auxin signaling [15]. In *A. thaliana*, WOX5 is expressed in the root apical meristem, wherein it promotes the division of stem cells [16]. Additionally, WOX9, which belongs to the intermediate clade, is reportedly involved in the development of flowers and embryos in many species [14]. In the monocotyledonous plant *Lilium lancifolium*, WOX9 and WOX11 control the formation of a reproductive organ (bulbil) via the cytokinin pathway [17].

*D. catenatum* is a perennial herb (family: Orchidaceae) that is valued as an ornamental and medicinal plant. Wild *D. catenatum*, which can withstand harsh environments, often grows slowly under semi-shade conditions on the rocky terrain in mountainous regions at an altitude of 1600 m. Because of the destruction of its habitat due to human activities, the availability of wild *D. catenatum* germplasm resources is decreasing [18,19]. In recent years, there have been many studies on *D. catenatum* growth and development and abiotic stress-related genes, such as the CIPK24 gene, the WRKY gene family, the MYB transcription factor superfamily, the APETALA2 (AP2) transcription factor, and the SnRK family [20,21,22,23,24]. Studies have also shown that the WOX genes of *D. catenatum* are widely expressed in the protocorm unique to orchids, indicating that WOX is involved in the growth and development of protocorm [25]. There have also been few studies on biological stress, such as *D. catenatum* enhancing resistance to pathogens through signal transduction of jasmonic acid (JA) [26]. This study identified and characterized the plant-specific WOX transcription factor genes in *D. catenatum* and also analyzed their physicochemical characteristics, phylogenetic relationships, and structures, as well as the conserved domains and subcellular localization of the encoded proteins. Moreover, unlike before [27], the phytohormone-signal-transduction-related cis-acting elements in the DcaWOX promoters and DcaWOX expression levels in response to different phytohormone treatments were analyzed in this study. The results of this study have laid the foundation for future studies on DcaWOX genes’ biological functions.

## 2. Materials and Methods

### 2.1. Plant Materials and Phytohormone Treatments

*D. catenatum* Lindl. plants were cultivated in a soil: sand mixture (5:1) in a greenhouse set at 24 °C with 70% relative humidity and a 16 h photoperiod. Plants that were growing uniformly after 6 months were treated with phytohormones for 3 or 6 h. When the plant is about 10 cm tall, soft green in color, tillerless, and exhibiting vigorous growth, it has not yet reached the flowering stage. The untreated plants served as controls. The samples were then flash-frozen in liquid nitrogen and stored at −80 °C until used.

### 2.2. Identification of WOX Family Genes in D. catenatum

The annotated genes and proteins in *D. catenatum* were obtained from the *D. catenatum* genome database (http://orchidbase.itps.ncku.edu.tw/est/Dendrobium_2019.aspx) (accessed on 12 December 2021). Additionally, 15 *A. thaliana* WOX genes were downloaded from the TAIR database (http://www.arabidopsis.org/) (accessed on 12 December 2021) (Appendix A). The *D. catenatum* WOX genes were identified on the basis of local BLAST searches and the hidden Markov model [28]. More specifically, a local database constructed for the DcaWOX proteins was searched using all 15 known *A. thaliana* WOX protein sequences to preliminarily identify candidate genes, which were verified using the hidden Markov model. Finally, the candidate genes were identified using Pfam (http://pfam.sanger.ac.uk) (accessed on 16 April 2022) (accession number PF00046). The SMART program was used to check for the presence of a conserved homeodomain. The sequences with WOX functional domains were retained for the following analysis [29].

### 2.3. Sequence Characterization and Phylogenetic Analysis

The sequences of the identified DcaWOX proteins were characterized using the online tool Expasy (https://web.expasy.org/protparam/) (accessed on 20 April 2022) [30]. Moreover, the subcellular localization of the proteins was predicted using Plant-mPLoc (http://www.csbio.sjtu.edu.cn/bioinf/plant-multi/) (accessed on 7 May 2022). The *D. catenatum* WOX proteins were aligned with the *A. thaliana* WOX protein sequences using the MEGA 7.0 software (Nei Masatoshi, United states). Furthermore, the neighbor-joining method was used to construct a phylogenetic tree with 1000 bootstrap replicates to assess its reliability [31].

### 2.4. Gene Structure and Conserved Motif Analyses

The conserved homeodomain encoded in the DcaWOX genes was identified using MEGA 7.0 and the GeneDoc software (Karl B. Nicholas, United states). The DcaWOX structures were analyzed on the basis of the annotation files in the *D. catenatum* genome database. The promoter regions 2.0 kb upstream of the initiation codon of the *D. catenatum* DcaWOX genes were analyzed and visualized using the PlantCARE online program (http://bioinformatics.psb.ugent.be/webtools/plantcare/html/) (accessed on 7 May 2022) [32].

The conserved domains encoded by the candidate genes were identified using SMART. Additionally, the conserved domains in the DcaWOX proteins were examined using MEME (https://meme-suite.org/meme/index.html) (accessed on 18 April 2022), with the number of motifs set to 20 and the width range set to 1–50 amino acids [33]. The three-dimensional structures of the DcaWOX proteins were predicted using SWISS-MODEL (https://swissmodel.expasy.org/) (accessed on 10 May 2022) [34].

### 2.5. Subcellular Localization

The DcaWOX2, 3a, 5, 9, 11b, and 13a coding sequences were amplified by PCR using specific primers and Phanta^®^ Max SuperFidelity DNA Polymerase (Vazyme Biotech Co., Ltd.; Nanjing, China). The amplified sequences were inserted into the pRI101-GFP vector using ClonExpress^®^ II (Vazyme Biotech Co., Ltd.; Nanjing, China). The 35S: GFP-DcaWOX constructs were inserted into Agrobacterium tumefaciens EHA105 cells via electroporation for the subsequent injection into *Nicotiana benthamiana* leaves. A laser confocal microscope was used to examine the tobacco leaves after a 3-day incubation in a greenhouse, as previously described [35].

### 2.6. Transient Expression in N. benthamiana

Transcriptional activation activities were analyzed using 35S: GFP-DcaWOX2, 3a, 5, 9, 11b, and 13a as the effectors and ProAtTPL:LUC as the reporter [36]. Briefly, 35S: GFP-DcaWOX2, 3a, 5, 9, 11b, and 13a were injected into *N. benthamiana* leaves along with ProAtTPL:LUC. The plants were then incubated in a greenhouse for 3 days before the leaves were sprayed with 1 mM fluorescein. After a 3 min incubation in darkness, the luciferase-related luminescence was observed using an automated chemiluminescence imaging system and photographed with a CCD camera [37]. The SPSS 22.0 (Norman H. Nie, C. Hadlai (Tex) Hull and Dale H. Bent, State of California) independent samples *t*-test was performed to analyze the integrated density (*n* = 5).

### 2.7. Transcriptomic Analysis

The transcriptome data used for analyzing the *DcaWOX* expression levels in various tissues were obtained from the *D. catenatum* genome database (http://orchidbase.itps.ncku.edu.tw/est/Dendrobium_2019.aspx) (accessed on 28 May 2022). The transcriptome data for phytohormone-treated *D. catenatum* plants were collected from the Biodiversity Data Center (https://data.iflora.cn/Home/DataContent?data_gd=89131009-82d9-ad4b-faaa-1bad087095e2) (accessed on 28 May 2022). The SPSS 22.0 independent samples *t*-test was performed to analyze the *DcaWOX* expression levels (*n* = 3), which were visualized in heat maps generated using Tbtools (Chengjie Chen, South China Agricultural University).

## 3. Results

### 3.1. Identification of WOX Family Genes in D. catenatum

The publication of the complete *D. catenatum* genome enabled the genome-wide identification of genes. We compared the protein sequences of 15 AtWOX transcription factor family members with the *D. catenatum* genome sequence. The identified candidates were validated by screening for conserved domains using Pfam and SMART. A total of 10 *DcaWOX* genes that included homeobox motif sequences were identified in the *D. catenatum* genome and then named according to their similarities to *A. thaliana WOX* sequences (Figure 1).

The DcaWOX proteins were divided into the WUS, intermediate, and ancient clades, which comprised five, three, and two members, respectively. Evolutionary developmental relationships suggested that DcaWOX2, 4, 5, and 9 lack homologs, whereas DcaWOX3, 11, and 13 have two homologs. The number of amino acids in the DcaWOX proteins ranged from 166 (DcaWOX4) to 337 (DcaWOX9), and the molecular weights ranged from 18.64 (DcaWOX4) to 36.91 (DcaWOX9) kDa. The theoretical isoelectric point (pI) was between 5.26 (DcaWOX13b) and 10.25 (DcaWOX4). Additionally, six of the DcaWOX proteins (60%) had a pI < 7, indicating that most of the DcaWOX proteins are rich in acidic amino acids. The calculated grand average of hydrophobicity values ranged from −0.88 (DcaWOX13b) to −0.23 (DcaWOX11b), reflecting the hydrophilicity of all DcaWOX proteins. Using an online tool, the DcaWOX proteins were predicted to be localized in the nucleus, wherein they may function as transcription factors (Table 1).

### 3.2. Analysis of D. catenatum WOX Phylogenetic Relationships, Genetic Structures, and Conserved Domains

The diversity of the *D. catenatum* DcaWOX transcription factor genes was studied on the basis of their structures as well as the structural characteristics and conserved motifs of the encoded proteins. The WOX protein family members reportedly contain a conserved homeodomain [38]. The DcaWOX protein sequences were aligned using MEGA 7.0, whereas the conserved domain was visualized using GeneDoc and TBtools [39]. The results indicated that most of the DcaWOX transcription factors are conserved and contain a “helix-loop-helix-turn-helix” homeodomain composed of 60 amino acids (Figure 2A). A phylogenetic tree was built for the *D. catenatum* WOX proteins using the MEGA 7.0 software (Figure 2B). Additionally, the *DcaWOX* promoters’ structure was analyzed using online tools (e.g., PlantCARE). The members of the same subclade generally contained the same number of introns, but the number of introns varied between clades, with two exons detected in the WUS clade genes, two or three exons in the intermediate clade genes, and three exons in the ancient clade genes (Figure 2C). Of the predicted cis-acting elements in the 2.0 kb region upstream of *DcaWOX* genes, 24 were related to responses to phytohormones, light, drought stress, and low temperatures. Moreover, the cis-acting elements varied among genes. More specifically, the phytohormone-related cis-acting elements were revealed to be associated with responses to Methyl Jasmonate, salicylic acid (SA), abscisic acid (ABA), gibberellin, and auxin. Each *DcaWOX* gene promoter contained at least two cis-acting elements, suggestive of their importance for plant adaptations to environmental stress (Figure 2D). These findings reflect the involvement of the WOX transcription factors in many important aspects of *D. catenatum* growth and development.

The analysis of the structural characteristics of *D. catenatum* DcaWOX transcription factors using SMART and MEME confirmed that all 10 candidates had a homeodomain (Figure 2E). Furthermore, 20 conserved motifs were identified in the 10 DcaWOX proteins (Motifs 1–20). Motifs 1 and 2 were detected in all DcaWOX transcription factors. With the exception of DcaWOX4, the WUS clade members contained Motif 5 (WUS-box) at the C-terminus. The intermediate clade proteins contained Motifs 3 and 8, whereas the ancient clade proteins included Motifs 4, 12, and 17. In each clade, the conserved motifs were essentially the same. Accordingly, the DcaWOX transcription factors belonging to the same clade may have similar biological functions. These findings also verify the reliability of the phylogenetic analysis (Figure 2F). To further analyze the protein structure of the DcaWOX family, SWISS-MODEL was used to predict the three-dimensional structure of the proteins. The resulting models included a “helix-loop-helix-turn-helix” structure. The DcaWOX transcription factors in each subclade had similar three-dimensional structures, indicative of a shared function. The observed structural diversity between subclades may reflect functional differences (Figure 2G).

### 3.3. Subcellular Localization

The predicted DcaWOX proteins are localized in the nucleus and may function as transcription factors that regulate plant growth and development. Selected *DcaWOX* genes from the WUS clade (DcaWOX2, DcaWOX3a, and DcaWOX5), the intermediate clade (DcaWOX9 and DcaWOX11b), and the ancient clade (DcaWOX13a) were analyzed further. The 35S: GFP-DcaWOX constructs were inserted into tobacco leaves. The subsequent examination involving DAPI revealed green fluorescence in the nucleus (Figure 3). This is consistent with the predicted results, suggesting that DcaWOX proteins may function as transcription factors.

### 3.4. DcaWOX Regulates Plant Cell Division and Differentiation by Activating TPL Expression

The WUS transcription factor can regulate the expression of TPL and some TPR genes in meristematic tissue, which is necessary for the maintenance of the meristem [40,41]. A previous study confirmed that WUS balances cell division and differentiation by modulating the expression of its downstream target genes [41]. In *A. thaliana* and other species, WUS activates the expression of the TPL gene through its WUS-box and EAR motif, thereby inhibiting the expression of the downstream target genes and maintaining the meristem [42,43,44]. Therefore, we hypothesized that the WUS clade transcription factors in *D. catenatum* regulate cell division and differentiation through the same mechanism. To test this hypothesis, the AtTPL promoter (a sequence 2.0 kb upstream of ATG) was analyzed to determine whether the WUS clade transcription factors can induce AtTPL expression. The observed LUC activity was greater in the samples co-transfected with the ProAtTPL:LUC reporter plasmid and the 35S:DcaWOX2, 35S:DcaWOX3a, and 35S:DcaWOX5 (i.e., WUS clade members) effector plasmids than in the samples lacking the effector plasmids. This suggests that DcaWOX2, DcaWOX3a, and DcaWOX5 can activate AtTPL transcription in vivo. In contrast, there were no significant differences in the LUC activities between the samples co-transfected with the ProAtTPL:LUC reporter plasmid and the 35S:DcaWOX9, 35S:DcaWOX11b, and 35S:DcaWOX13a effector plasmids and the samples with ProAtTPL:LUC alone, implying DcaWOX9, DcaWOX11b, and DcaWOX13a cannot induce the expression of AtTPL in vivo (Figure 4). Hence, the DcaWOX members of the WUS clade can activate AtTPL expression, whereas the members of the intermediate and ancient clades cannot.

### 3.5. WOX Gene Expression Profiles in D. catenatum

Gene expression patterns reflect gene functions to some extent [45]. To functionally characterize the *D. catenatum* WOX genes, we analyzed their expression in various tissues and in response to phytohormone treatments using transcriptome data. The *DcaWOX* genes were expressed in different tissues (Figure 5A), which was in accordance with the previously reported functions of WOX transcription factors. Specifically, DcaWOX2 and DcaWOX9 were highly expressed in the gynostemium unique to orchids, suggesting they may be species-specific genes. Both DcaWOX3a and DcaWOX3b were most highly expressed in flower buds, indicating they may contribute to floral development. In contrast, DcaWOX4 and DcaWOX5 were mainly expressed in the roots, with DcaWOX5 highly expressed in the root tip. Thus, these two genes may be involved in root development. 

Although the DcaWOX11a expression level was highest in the green root tip, the expression of its homolog, DcaWOX11b, peaked in the pollinium, suggesting the divergent evolution of these genes. Similarly, DcaWOX13a was mainly expressed in flower buds, and DcaWOX13b was primarily expressed in the pollinium, suggesting that these genes may be associated with the development of different flower parts. The gene expression analysis indicated that *WOX* genes are widely expressed in the roots, stems, leaves, and flowers in *D. catenatum*, implying that the encoded transcription factors play crucial roles in *D. catenatum* development and growth.

After a 6 h indole−3−acetic acid (IAA) treatment, the DcaWOX5 expression level gradually increased (Figure 5B). Additionally, the expression of DcaWOX9 (intermediate clade) was highest in response to a 3 h SA treatment. Similarly, DcaWOX13a was also highly expressed following the SA treatment. These results are consistent with the predicted phytohormone-associated cis-acting elements in the gene promoters. The phytohormone treatments differentially affected the expression patterns of DcaWOX3a, DcaWOX3b, DcaWOX11a, DcaWOX11b, DcaWOX13a, and DcaWOX13b, which may reflect the functional diversification of the DcaWOX3, 11, and 13 homologs during evolution.

## 4. Discussion

*D. catenatum* is a traditional herb that produces flavonoids, polysaccharides, and other metabolites that are often used in clinical research, making it an economically valuable medicinal plant species [46,47]. Earlier research revealed that WOX transcription factors are crucial for plant growth and development because they maintain meristem stem cells [43,48,49]. However, there are relatively few reports describing DcaWOX genes and, in particular, their expression patterns in response to various phytohormones. Consistent with previous research, we have seen in Orchid 10 WOX transcription factor genes identified in *D. Catenatum* [27]. The encoded protein sequences, along with 10 *A. thaliana* WOX protein sequences, were used to construct a phylogenetic tree (Figure 1) and analyze the phylogenetic relationships among the DcaWOX genes (Figure 2B). To further characterize the DcaWOX transcription factors, their gene structures (Figure 2C,D) and the conserved motifs in the encoded proteins (Figure 2A,E–G) were analyzed. The results indicated the DcaWOX proteins in the same subclade had similar motifs, suggestive of functional redundancies among members of the same subclade.

To elucidate the DcaWOX functions, we selected *DcaWOX2*, *3a*, *5*, *9*, *11b*, and *13a* from the three clades for the subsequent analysis. The subcellular localization analysis involving tobacco leaves demonstrated that the encoded proteins were present in the nucleus (Figure 3), consistent with the predictions. Moreover, the proteins may function as transcription factors mediating numerous processes that regulate plant growth and development. Previous studies confirmed that the WUS transcription factor containing the WUS-box and EAR domains in *A. thaliana* can bind directly to the TPL promoter and activate transcription to regulate cell division. Accordingly, we examined whether other members of the WUS clade with WUS-box and EAR domains were functionally similar to the corresponding proteins in *A. thaliana*. In our study, DcaWOX2, 3a, and 5 (WUS clade), which contain the WUS-box and EAR domains, induced TPL transcription, whereas DcaWOX9, 11b, and 13a, which lack these domains, did not activate TPL expression (Figure 4). These findings are in accordance with the results of earlier research on *A. thaliana.* Hence, the structures and functions of the WUS clade members appear to be highly conserved in the monocotyledonous species *D. catenatum* and the dicotyledonous species *A. thaliana*.

Plant growth and development are regulated by a variety of signaling factors (e.g., phytohormones) that modulate plant defenses and stress resistance. The WOX transcription factors are reportedly involved in phytohormone signaling pathways in some species. For example, they can affect rice root cell division via their effects on the auxin and cytokinin signaling pathways [50,51]. The effects of phytohormones on DcaWOX expression were determined on the basis of transcriptome data for samples treated with diverse phytohormones (i.e., ABA, IAA, JA, and SA). The data indicated that the DcaWOX genes were differentially expressed following the phytohormone treatments (Figure 5B). Existing studies have shown that *D. catenatum* increases resistance to the necrotrophic Southern Blight pathogen after the application of the exogenous hormone Methyl Jasmonate (MeJA) [26]. In this study, DcaWOX11 and DcaWOX13 were expressed at 3 or 6 h under JA treatment, suggesting that DcaWOX may also have a disease-resistant effect through the JA pathway. Thus, these WOX genes help regulate various developmental processes, but the mechanism underlying their Involvement in various phytohormone-related responses needs to be further studied.

## 5. Conclusions

In this study, we performed a genome-wide analysis of the WOX gene family in *D. catenatum*, which resulted in the identification of 10 DcaWOX genes. The structural and functional differences among the WUS, intermediate, and ancient clade genes were revealed. Additionally, the phytohormone-induced expression patterns for the 10 DcaWOX genes were characterized. The results of this study provide the basis for an in-depth study of the molecular functions of the DcaWOX transcription factors.

## Figures and Tables

**Figure 1 genes-13-01481-f001:**
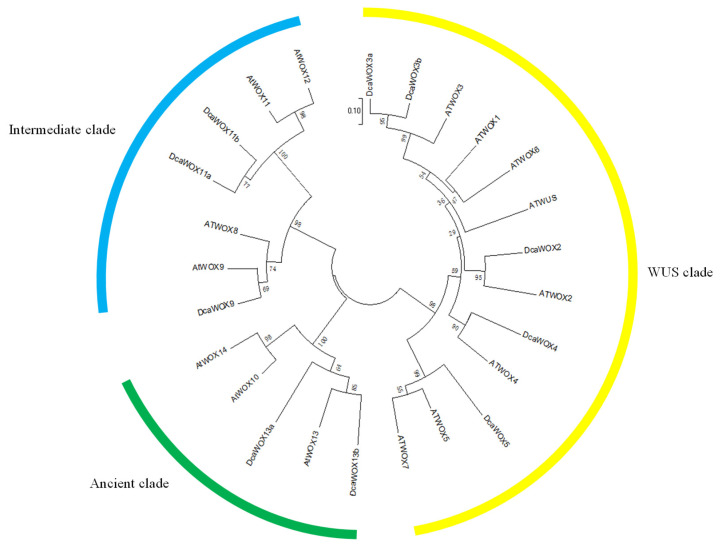
Phylogenetic tree of WOX proteins from *D. catenatum* and *A. thaliana*. The tree was constructed according to the neighbor-joining method implemented using MEGA 7.0 software and a bootstrap analysis with 1000 iterations.

**Figure 2 genes-13-01481-f002:**
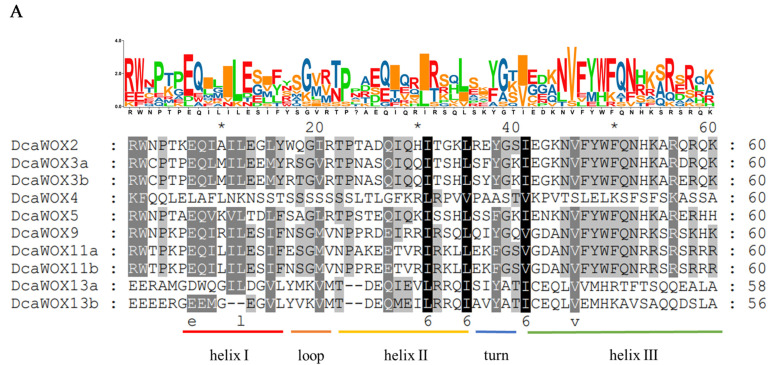
*D. catenatum WOX* gene structures and the conserved motifs in the encoded proteins. (**A**) Alignment of the homeodomain. (* Indicates the separation of every ten amino acid residues) (**B**) Phylogenetic tree of *D. catenatum* WOX proteins. (**C**) *DcaWOX* gene structure. (**D**) Cis-acting elements in the *DcaWOX* gene promoters. (**E**) Homeodomain revealed by SMART. (**F**) Conserved motifs in *D. catenatum* WOX proteins identified by MEME. Each motif is indicated by a specific color. (**G**) Three-dimensional conformation of *D. catenatum* WOX proteins.

**Figure 3 genes-13-01481-f003:**
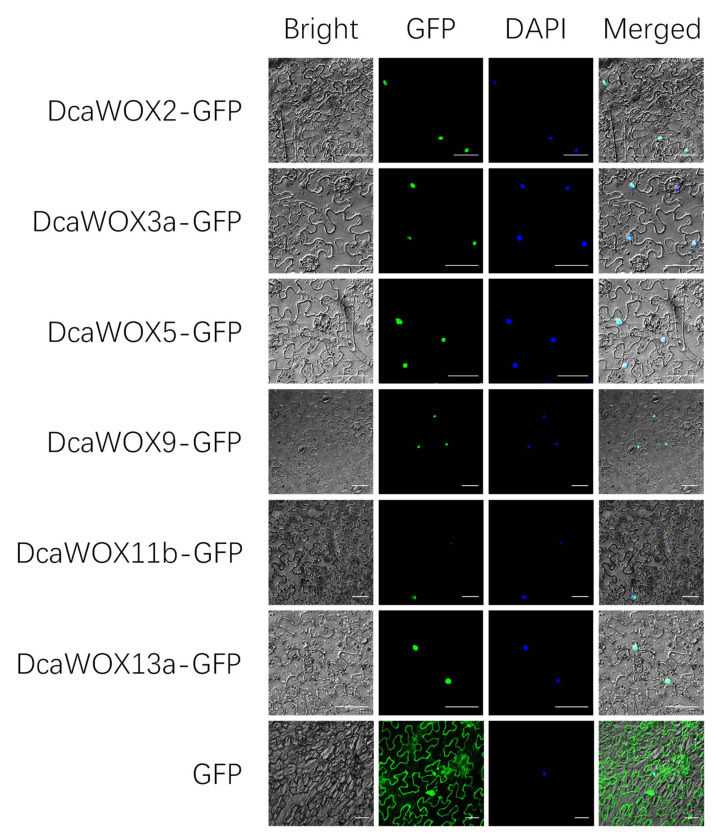
Subcellular localization of DcaWOX-GFP in *Nicotiana benthamiana* leaves. DcaWOX2-GFP, DcaWOX3a-GFP, DcaWOX5-GFP, DcaWOX9-GFP, DcaWOX11b, and DcaWOX13a were localized in the nucleus. Bar = 50 μm.

**Figure 4 genes-13-01481-f004:**
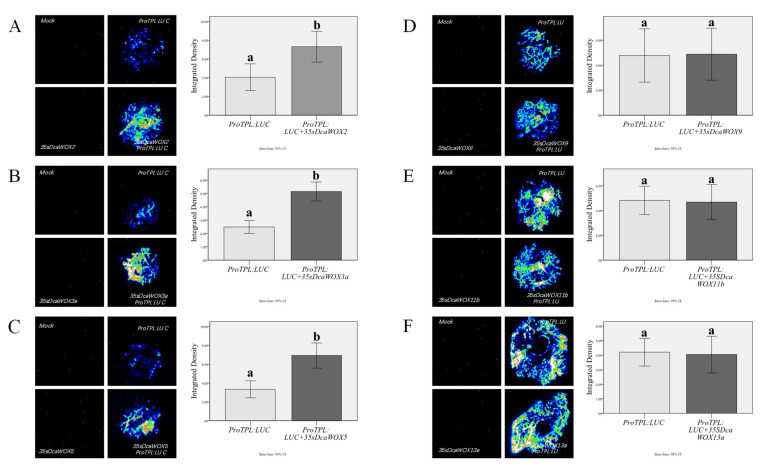
Analysis of the effects of DcaWOX2, DcaWOX3a, DcaWOX5, DcaWOX9, DcaWOX11b, and DcaWOX13a. *TPL* expression was activated by (**A**) DcaWOX2, (**B**) DcaWOX3a, and (**C**) DcaWOX5. *TPL* expression was not activated by (**D**) DcaWOX9, (**E**) DcaWOX11b, and (**F**) DcaWOX13a. a, b indicates significant differences among treatments (*n* = 5).

**Figure 5 genes-13-01481-f005:**
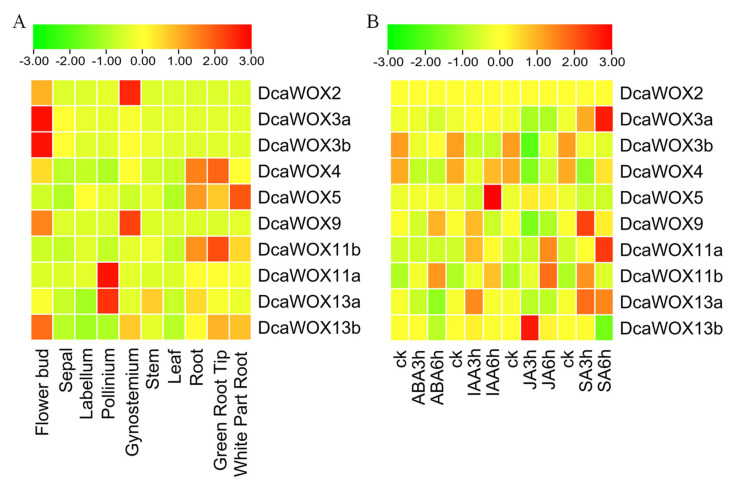
Expression profiles of *D. catenatum WOX* genes. (**A**) *DcaWOX* genes expression in different tissues. (**B**) *DcaWOX* genes expression after treatments with ABA, IAA, JA, and SA. ABA: abscisic acid; IAA: indole−3−acetic acid; JA: jasmonic acid; SA: salicylic acid; ck: control check; 3 h and 6 h: time-points after initiating the phytohormone treatments. (*n* = 3).

**Table 1 genes-13-01481-t001:** Identification and characteristics of *WOX* genes in *D. catenatum*.

Gene Name	Accession Number	CDS Length (bp)	Protein Size (aa)	MW (kD)	PI	GRAVY	Protein Localization
DcaWOX2	XP_020689224.1	696	231	26.02	6.66	−0.68	Nucleus
DcaWOX3a	XP_020698032.1	579	192	21.65	9.11	−0.70	Nucleus
DcaWOX3b	XP_020698705.1	645	214	24.34	6.79	−0.83	Nucleus
DcaWOX4	XP_028548805.1	501	166	18.64	10.25	−0.64	Nucleus
DcaWOX5	XP_020673683.1	555	184	21.26	8.85	−0.72	Nucleus
DcaWOX9	XP_028553243.1	1014	337	36.91	7.79	−0.26	Nucleus
DcaWOX11a	XP_020695082.1	789	262	27.81	5.78	−0.23	Nucleus
DcaWOX11b	XP_028555924.1	762	253	27.33	6.73	−0.24	Nucleus
DcaWOX13a	XP_020676225.2	657	218	25.16	6.98	−0.68	Nucleus
DcaWOX13b	XP_020685087.1	780	259	29.92	5.26	−0.88	Nucleus

## Data Availability

Not applicable.

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
