# Peer review of "Genome-Wide Analysis of the WOX Transcription Factor Genes in Dendrobium catenatum Lindl."

_genes, 2022, doi:10.3390/genes13081481_

Round 1

Reviewer 1 Report

1. The authors conduct a genome-wide analysis for WOX transcription factor genes in a popular orchid species, Dendrobium catenatum Lindl. The research is complete and provides some value for the field of orchid research. It needs minor revision before publication.

2. Figure 4: The letters that present significant differences are too small and thus can not recognize which word they are. The authors need to improve them.

3. M&M: The morphology, physiology, and age of plant materials should be described in detail or the authors could provide iconic photos for it.

4. A section on statistics should be added and the authors need to provide information about how many replicates were conducted in each treatment, particularly for the part of gene expression.

5. Avoid using first-person writing throughout the manuscript.

6. The authors need to discuss the present findings by comparing them with the results of a previous report, Thakku R. Ramkumar, Madhvi Kanchan, Santosh Kumar Upadhyay, Jaspreet K. Sembi, Identification and characterization of WUSCHEL-related homeobox (WOX) gene family in economically important orchid species Phalaenopsis equestris and Dendrobium catenatum, Plant Gene, Volume 14, 2018, Pages 37-45.

7. The authors need to enrich the introduction by adding some content for describing orchid genetics, Dendrobium gene regulation, or something like that.

8. The authors could enrich their discussion by organizing a conclusive graph for the possible key function network of WOX transcription factor genes.

9. L166: were - was

10. Figure 2: All graphs are too small.

Reviewer 2 Report

Dear Author,

The manuscript entitled “Genome-wide analysis of the WOX transcription factor genes in Dendrobium catenatum Lindl.” was revised. The Manuscript can be published after doing some minor revisions and suggestions. However, the reference list requires major revision. The entire list should be reconsidered and rewritten in the light of the journal's writing rules. The discussion section should be expanded further and the data should be supplemented by citing other sources. The discussion section also remains as the weakest part of the article. This suggestions and revisions described in attached file

Kind Regards

Author Response

This manuscript is a resubmission of an earlier submission. The following is a list of the peer review reports and author responses from that submission.